# Generation and deposition of Aβ43 by the virtually inactive presenilin-1 L435F mutant contradicts the presenilin loss-of-function hypothesis of Alzheimer's disease

Benedikt Kretner[1,2,†], Johannes Trambauer[1,†], Akio Fukumori[2], Janina Mielke[3], Peer-Hendrik Kuhn[2,4], Elisabeth Kremmer[5,6], Armin Giese[3], Stefan F Lichtenthaler[2,4,5], Christian Haass[1,2,5], Thomas Arzberger[2,3,7] & Harald Steiner[1,2,*]

## Abstract

As stated by the prevailing amyloid cascade hypothesis, Alzheimer's disease (AD) is caused by the aggregation and cerebral deposition of long amyloid-β peptide (Aβ) species, which are released from a C-terminal amyloid precursor protein fragment by γ-secretase. Mutations in its catalytic subunit presenilin-1 (PS1) increase the Aβ42 to Aβ40 ratio and are the major cause of familial AD (FAD). An opposing hypothesis states that loss of essential presenilin functions underlies the disease. A major argument for this hypothesis is the observation that the nearly inactive PS1 L435F mutant, paradoxically, causes FAD. We now show that the very little Aβ generated by PS1 L435F consists primarily of Aβ43, a highly amyloidogenic species which was overlooked in previous studies of this mutant. We further demonstrate that the generation of Aβ43 is not due to a trans-dominant effect of this mutant on WT presenilin. Furthermore, we found Aβ43-containing plaques in brains of patients with this mutation. The aberrant generation of Aβ43 by this particular mutant provides a direct objection against the presenilin hypothesis.

**Keywords** Alzheimer's disease; amyloid-β peptide 43; γ-secretase; neurodegeneration; presenilin

**Subject Category** Neuroscience

## Introduction

According to the widely believed amyloid cascade hypothesis, Alzheimer's disease (AD) is triggered by the pathogenic accumulation of long amyloid-β peptide (Aβ) species such as Aβ42 in the brain of affected patients. Aβ is derived from sequential processing of the β-amyloid precursor protein (APP) by β- and γ-secretase (Lichtenthaler *et al*, 2011). The latter cleavage starts at the ε-site within the APP transmembrane domain and following a series of carboxy-terminal trimming steps ultimately releases two major Aβ forms, Aβ40 as principal product as well as smaller amounts of Aβ42 (Lichtenthaler *et al*, 2011). Due to its higher hydrophobicity, the slightly longer Aβ42 peptide is more prone to aggregation than Aβ40, a property, which likely underlies its neurotoxicity (Haass & Selkoe, 2007). The amyloid cascade hypothesis is strongly supported by genetic evidence from familial AD (FAD) cases showing that most mutations in APP as well as in presenilin-1 (PS1) and its homologue presenilin-2 (PS2), which constitute the catalytic subunit of γ-secretase (Lichtenthaler *et al*, 2011), cause an increased ratio of Aβ42 to Aβ40 (Scheuner *et al*, 1996), i.e. that FAD mutations are associated with substrate and protease. Moreover, a mutation in immediate vicinity to the β-secretase cleavage site in APP has recently been identified in the Icelandic population that protects against AD (Jonsson *et al*, 2012), while a double mutation at this site occurring in a Swedish family (swAPP) results in an exceptionally strong increase of Aβ generation (Citron *et al*, 1992).

Since overproduction of Aβ42 in APP transgenic mice failed to induce neurodegeneration despite severe amyloidosis (Irizarry *et al*, 1997) and because many PS1 and PS2 FAD mutations strongly

---

1 Biomedical Center, Metabolic Biochemistry, Ludwig-Maximilians-University Munich, Munich, Germany
2 DZNE - German Center for Neurodegenerative Diseases, Munich, Germany
3 Center for Neuropathology and Prion Research, Ludwig-Maximilians-University Munich, Munich, Germany
4 Neuroproteomics, Klinikum rechts der Isar and Institute for Advanced Study, Technische Universität München, Munich, Germany
5 Munich Cluster for Systems Neurology (SyNergy), Munich, Germany
6 Institute of Molecular Immunology, Helmholtz Zentrum München, Munich, Germany
7 Department of Psychiatry and Psychotherapy, Ludwig-Maximilians-University Munich, Munich, Germany
*Corresponding author: Tel: +49 89 4400 46535; Fax: +49 89 4400 46546; E-mail: harald.steiner@med.uni-muenchen.de
†These authors contributed equally to this work

impair processing of γ-secretase substrates such as Notch (Weggen & Beher, 2012), an alternative hypothesis for the disease mechanism of AD, the "presenilin hypothesis", has been suggested, which states that AD is caused by a loss of essential presenilin functions (Shen & Kelleher, 2007). This has the important implication that therapeutic inhibition of γ-secretase to lower Aβ will not be beneficial in AD. Rather, it will be important to maintain normal presenilin function. The presenilin hypothesis was initially based on the observation that conditional PS knockout mice display classical features of neurodegeneration that also occur in AD (Saura *et al*, 2004). It received further support when a remarkable FAD mutation, PS1

L435F, was identified (Heilig *et al*, 2010). Biochemical analyses using cell culture models and recently also knockin mice revealed that this FAD mutant did not support presenilin endoproteolysis, an autocatalytic process to activate γ-secretase (Lichtenthaler *et al*, 2011), and nearly completely abrogated γ-secretase activity toward its substrates APP and Notch1 (Heilig *et al*, 2010, 2013; Xia *et al*, 2015). Thus, PS1 L435F represents a highly unusual PS1 FAD mutation that supports the concept that loss of presenilin function constitutes an essential trigger of early onset dementia, probably being more important than an elevated ratio of Aβ42/Aβ40 (Shen & Kelleher, 2007).

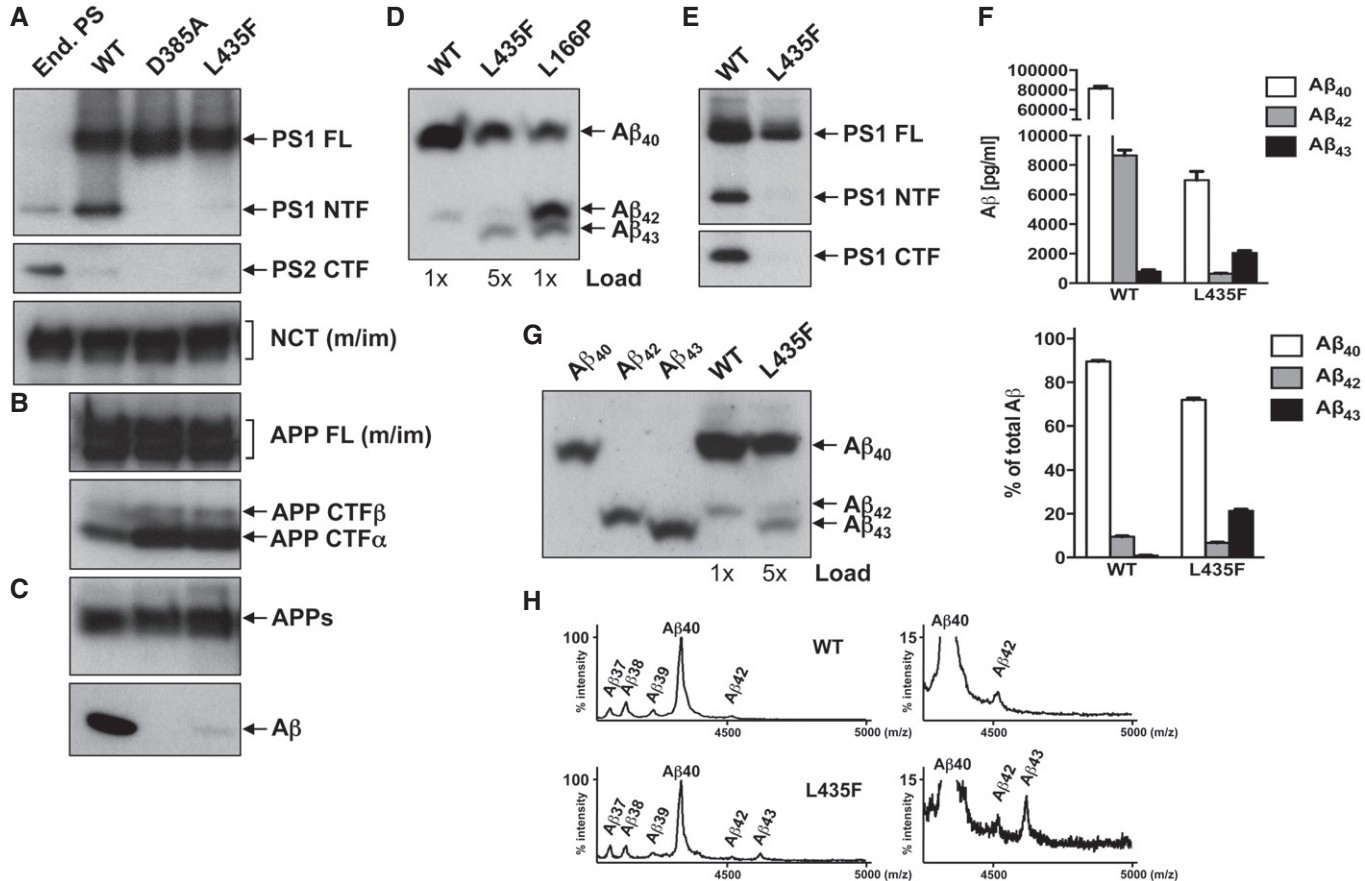

**Figure 1. The PS1 L435F FAD mutant strongly impairs APP processing while generating Aβ43.**

Pooled clones (A–D) and single-cell clones (E–H) of HEK293/sw cells untransfected or stably transfected with the indicated WT and mutant PS1 constructs were analyzed for γ-secretase expression and APP processing.

A  PS1, PS2, and NCT were analyzed in cell lysates by immunoblotting using antibodies PS1N (PS1), BI-HF5C (PS2), and N1660 (NCT), respectively.

B  Full-length APP and APP CTFs were analyzed by immunoblotting using antibody 6687.

C  Conditioned media were analyzed for secreted APPs by immunoblotting using antibody 22C11 and for total Aβ by combined immunoprecipitation/immunoblotting using antibodies 3552/2D8.

D  Conditioned media were analyzed for individual Aβ species on Tris-Bicine-Urea SDS-PAGE by combined immunoprecipitation/immunoblotting of Aβ using antibodies 3552/2D8. Pooled clones of HEK293/sw cells stably transfected with PS1 L166P were used as reference. Note that more sample was loaded for the PS1 L435F mutant to facilitate analysis.

E  PS1 expression levels were analyzed in cell lysates of HEK293/sw cells stably expressing PS1 WT or PS1 L435F by immunoblotting using antibodies PS1NT and 5E12, respectively.

F  Conditioned media were analyzed by ELISA specific for Aβ40, Aβ42, and Aβ43. Data represent mean ± s.e.m. (n = 6). Absolute levels and Aβ ratios are shown.

G  Secreted Aβ was analyzed as in (D). To verify individual Aβ species, Aβ standards were co-migrated.

H  Total Aβ in conditioned media was analyzed by MALDI-TOF MS following immunoprecipitation with antibody 4G8. Observed (Aβ42, 4513.6; Aβ43, 4615.3) and predicted molecular masses (Aβ42, 4514.1; Aβ43, 4615.2) were in good agreement.

Source data are available online for this figure.

## Results and Discussion

In a previous screen for mutations defective in presenilin autoproteolysis, two PS1 loss-of-function mutations, R278I, a known FAD mutation (Godbolt *et al*, 2004), and L435H, a synthetic mutation of L435, which severely reduced the processing of APP and Notch1 were described (Nakaya *et al*, 2005). Intriguingly, mass spectrometry (MS) analysis revealed that both PS1 R278I and L435H gave rise to abnormally high, nearly exclusive generation of Aβ43 (Nakaya *et al*, 2005), a finding that has been recapitulated *in vivo* for PS1 R278I knockin mouse models (Saito *et al*, 2011). Although Aβ43 is secreted only in very minor amounts under physiological conditions (Page *et al*, 2008; Saito *et al*, 2011), it is found in Aβ plaques in sporadic and FAD brain (Iizuka *et al*, 1995; Parvathy *et al*, 2001; Welander *et al*, 2009; Saito *et al*, 2011; Sandebring *et al*, 2013). Like Aβ42, it is highly neurotoxic, although the extent of toxicity compared to Aβ42 varies among assays (Saito *et al*, 2011; Burnouf *et al*, 2015; Meng *et al*, 2015). Aβ43 is deposited earlier than Aβ42 in mouse models of AD indicating that Aβ43 is a potent and probably also a primary nucleation factor of Aβ aggregates *in vivo* (Saito

*et al*, 2011; Zou *et al*, 2013). Taken together, this indicated that, similarly to the PS1 R278I or the L435H mutants, also PS1 L435F might secrete Aβ43, a property that was not investigated in previous studies (Heilig *et al*, 2010, 2013; Xia *et al*, 2015).

To investigate this possibility, we stably transfected HEK293 cells expressing swAPP (HEK293/sw) with cDNA encoding the PS1 L435F mutant or, as controls, PS1 WT and the catalytically inactive PS1 D385A mutant. As shown in Fig 1A, the constructs were robustly expressed in pooled clones and underwent normal γ-secretase complex formation as demonstrated by the replacement of endogenous PS2 and the maturation of nicastrin (NCT). We confirmed the previously reported loss-of-function phenotype for PS1 L435F showing virtually absence of presenilin endoproteolysis (Fig 1A), accumulation of uncleaved APP C-terminal fragments (CTFs) comparable to the catalytic inactive PS1 D385A mutation (Fig 1B) and nearly no Aβ total production (Fig 1C). Next, we investigated the Aβ profiles generated by these mutants. Tris-Bicine-Urea SDS-PAGE analysis, which allows electrophoretic separation of individual Aβ species (Wiltfang *et al*, 1997), showed a band for PS1 L435F that migrated faster than that of Aβ42 for PS1 WT and at a

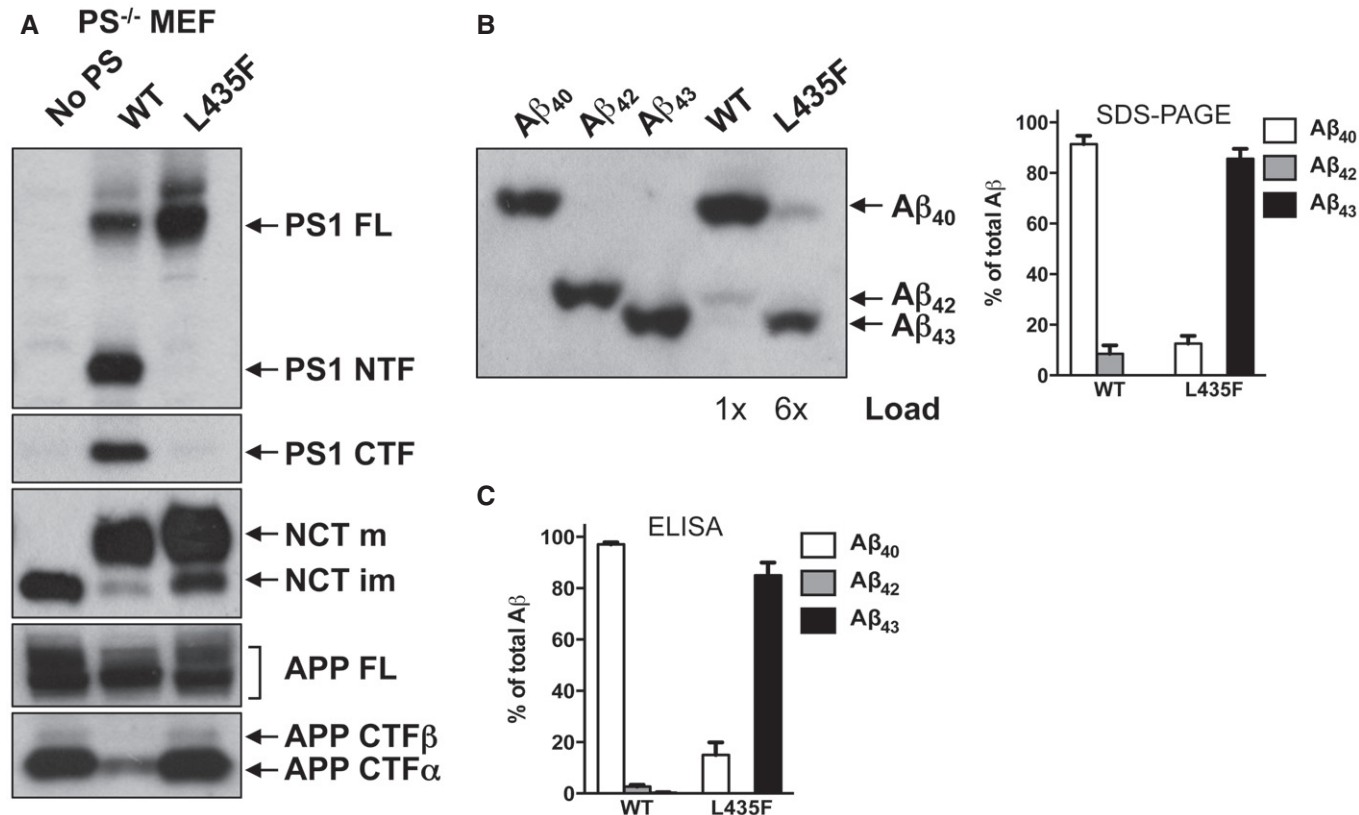

**Figure 2. The PS1 L435F FAD mutant generates Aβ43 as the predominant long Aβ species independent of PS1 WT.**

A  PS1/2$^{-/-}$ MEF cells stably transduced with PS1 WT or PS1 L435F were analyzed for PS1 expression and APP processing by immunoblotting as in Fig 1A. Antibody 5E12 was used for the detection of the PS1 CTF.

B  Stably transduced PS1/2$^{-/-}$ MEF cells of A were transiently transfected with APPsw-6myc and conditioned media were analyzed for Aβ species by Tris-Bicine-Urea SDS-PAGE. Individual Aβ species were verified by co-migration with Aβ standards (left panel) and quantified (right panel). Data represent mean ± s.e.m. of *n* = 3 independently performed transfections. Note that more sample was loaded for the PS1 L435F mutant to facilitate immunoblot analysis.

C  Secreted Aβ from cells in B was quantified by ELISA. Data represent mean ± s.e.m. of *n* = 5 (PS1 WT) or 6 (PS1 L435F) independently performed transfections.

Source data are available online for this figure.

comparable height of the Aβ43 band observed for the PS1 L166P mutant that was included in this analysis as a reference FAD mutant that besides high amounts of Aβ42 also generates Aβ43 (Moehlmann *et al*, 2002; Page *et al*, 2008) (Fig 1D). These data strongly suggests that PS1 L435F secretes Aβ43 and in even higher amounts than Aβ42 (Fig 1D). To further confirm these data, single-cell clones stably overexpressing PS1 WT or PS1 L435F (Fig 1E) were used next for a quantitative analysis of Aβ species using a previously described

highly sensitive Aβ43-specific ELISA (Saito *et al*, 2011). As shown in Fig 1F, Aβ43 was identified in the PS1 L435F mutant as prominent and more abundant species than Aβ42. Tris-Bicine-Urea SDS-PAGE analysis of the Aβ species co-migrated with standard peptides (Fig 1G) as well as MS analysis (Fig 1H) confirmed this result. Finally, consistent with the strong loss of function in APP processing, generation of the APP intracellular domain (AICD) was nearly abolished by the mutant (Fig EV1A). Thus, the loss-of-function

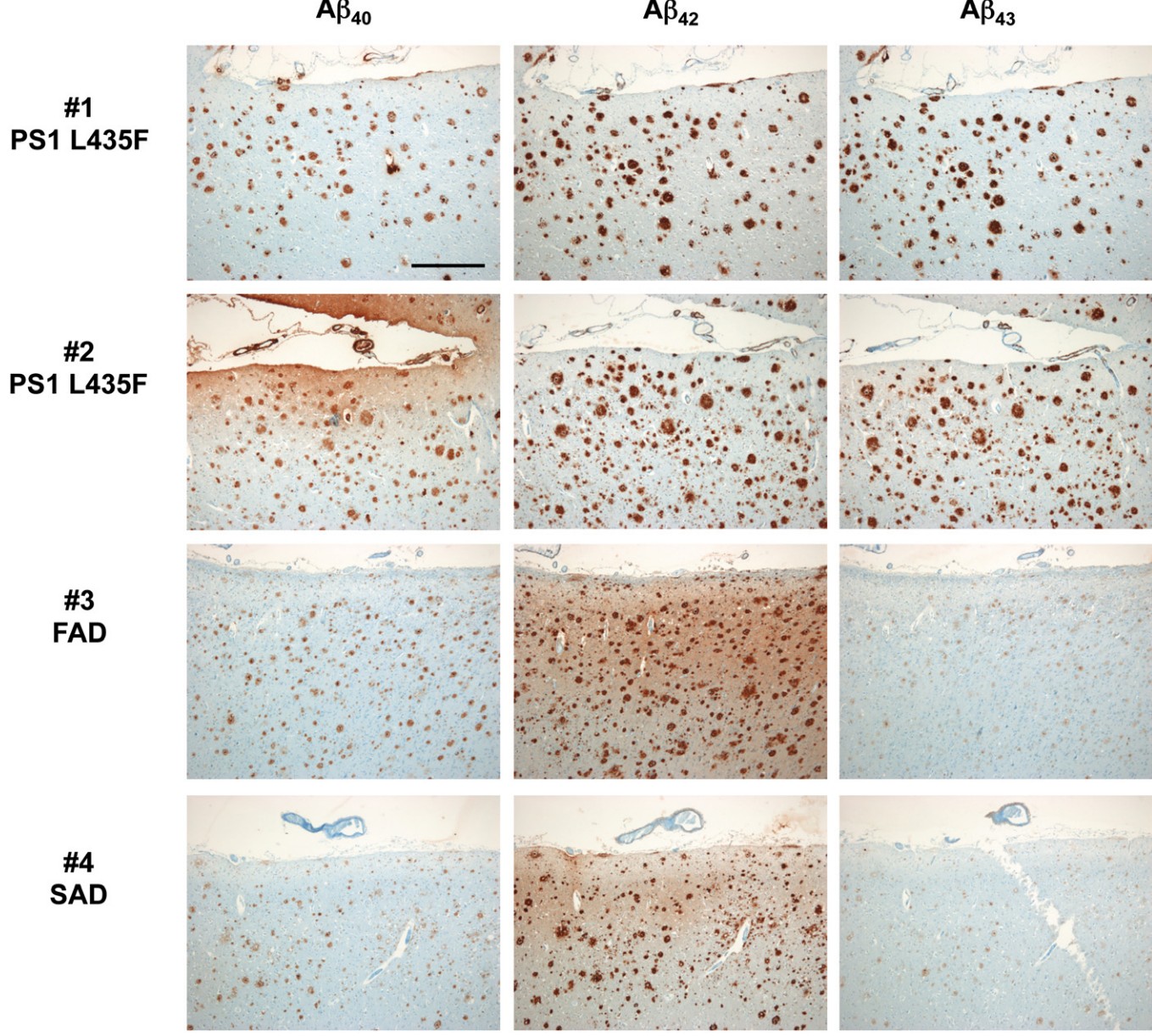

**Figure 3. Deposition of Aβ isoforms in the frontal cortex of AD cases with and without PS1 L435F mutation.**
Immunohistochemical detection of Aβ40 (left column), Aβ42 (medium column), and Aβ43 (right column) in consecutive frontal cortex paraffin sections of two FAD cases with PS1 L435F mutation (cases #1 and #2, 1st and 2nd rows), another FAD case with different PS1 mutation (case #3, 3rd row), and one SAD case (case #4, fourth row). In both PS1 L435F mutation cases, the numerous Aβ plaques mainly contain Aβ42 and Aβ43 but less Aβ40. This is in contrast to Aβ plaques of the control FAD and SAD cases, in which solely Aβ42 predominates and Aβ43 is sparse, even levels of Aβ40 seem to be lower than in the PS1 L435F cases. Note that in both PS1 L435F cases, several plaques are larger than those seen in cases #3 and #4, representing cotton wool plaques. Scale bar = 500 μm. Magnification is identical in all pictures.

phenotype of PS1 L435F is associated with an abnormal relative overproduction of Aβ43.

To explain the apparent paradox that PS1 L435F causes amyloid plaque deposition and AD in mutation carriers despite a near complete functional loss, it was suggested that the mutant protein inhibits and alters the catalytic activity of PS1 WT by a trans-dominant effect to stimulate generation of Aβ42 by PS1 WT (Heilig et al, 2013). To exclude the possibility that Aβ43 generation resulted from residual PS1 WT activity trans-dominantly disturbed by PS1 L435F, PS1/2$^{-/-}$ double-knockout mouse embryonic fibroblast (MEF) cells stably transduced with PS1 L435F, i.e. with the mutant as sole presenilin present in the cell, were analyzed next. As expected, the PS1 L435F mutant was hardly endoproteolysed, caused a strong loss of function of γ-secretase activity as judged from the accumulation of endogenous mouse APP CTFs similar to that in the parental PS1/2$^{-/-}$ MEFs (Fig 2A) and almost completely blocked AICD formation (Fig EV1B and C). As shown by Tris-Bicine-Urea SDS-PAGE analysis, Aβ43 became the major Aβ species secreted by the mutant upon transient overexpression of human APPsw-6myc (Kretner et al, 2013) (Fig 2B)—being generated in even higher relative amounts than in the HEK293/sw single-cell clone expressing the mutant (Fig 1G). Additional quantitative analysis by ELISA further confirmed this result and showed again that Aβ43 was the predominant Aβ species by the mutant when it was expressed in PS1/2$^{-/-}$ MEF cells (Fig 2C). Since these data demonstrate that the generation of Aβ43 by PS1 L435F is an intrinsic property of this mutant and not due to a trans-dominant effect on a dimeric PS1 WT/L435F complex, we thus finally revisited the human PS1 L435F FAD cases and asked whether Aβ43 was deposited in the brain of the affected two siblings analyzed previously (Heilig et al, 2010). Sequencing of genomic DNA isolated from frozen frontal cortex confirmed the presence of the PS1 L435F mutation in these cases and its absence in a control FAD case with a different PS1 mutation (Fig EV2). As shown in Figs 3 and EV3, substantial Aβ43 deposition was found in frontal cortex and hippocampus of both PS1 L435F cases. The staining intensity of Aβ43-positive plaques was much higher in the PS1 L435F mutation cases compared to the FAD case with a different PS1 mutation and an additional sporadic AD (SAD) case reaching the staining intensity levels of Aβ42, which is considered to be the major compound of Aβ plaques. Consistent with the previous report (Heilig et al, 2010), cotton wool plaques were observed in the two PS1 L435F mutation cases (Fig EV4A and B) that were frequently associated with neuritic pathology (Fig EV4C and D).

Taken together, although the PS1 L435F mutant leads to a near complete loss-of-function phenotype with only very little residual catalytic activity, we demonstrate that the mutant secretes substantial amounts of Aβ43 in cell culture models, higher than that of Aβ42. In line with these data, our neuropathological analysis showed that Aβ43 is deposited in high amounts in the brain of the mutation carriers. As evident from the nearly exclusive Aβ43 secretion by PS1 L435F in the absence of PS1 WT, the generation of pathogenic Aβ species can, however, not result from PS1 WT activity disturbed by the heterozygous expression of PS1 L435F as recently proposed (Heilig et al, 2013). Overall, the PS1 L435F is very similar to the PS1 R278I FAD mutation, for which a strong loss-of-function phenotype is associated with nearly exclusive secretion of Aβ43 (Nakaya et al, 2005; Saito et al, 2011). Homozygosity of PS1

R278I is embryonic lethal in mice; however, heterozygous mice (PS1 WT/R278I) are viable and develop plaque pathology associated with memory impairment when crossed with APP transgenic mice (Saito et al, 2011). Thus, analogous to PS1 R278I, the PS1 L435F mutation might induce FAD by Aβ43 production rather than by a loss of γ-secretase function as proposed previously. Indeed, considering that an increase of the Aβ42/Aβ40 ratio is sufficient to accelerate formation and stabilization of toxic oligomeric Aβ species even when the absolute Aβ concentration is very low (Kuperstein et al, 2010), it becomes clear that severe loss-of-function mutations that almost completely abolish total Aβ generation could cause FAD when they preferentially cause generation of Aβ43 rather than Aβ42 and Aβ40. Our results thus add to other criticisms raised against the presenilin hypothesis. For example, not all PS1 FAD mutations are associated with reduced γ-secretase activity (Weggen & Beher, 2012) and mutations in PS1, NCT, and PEN-2 that fully abolish γ-secretase function are associated with familial acne inversa, but not with dementia (Wang et al, 2010). The presenilin hypothesis can also not sufficiently explain APP FAD mutations and is inconsistent with a protective APP mutation in the Icelandic population, which reduces the risk for AD by reducing total Aβ levels (Jonsson et al, 2012). In conclusion, our data suggest that the essential and primary trigger of FAD in mutations like PS1 L435F is the secretion of Aβ43 arguing against the presenilin hypothesis.

# Materials and Methods

### Antibodies

Antibodies to PS1 N-terminal (PS1N, Capell et al, 1997, immunoblot (IB) 1:1,000) and PS2 C-terminal fragments (BI.HF5c, Steiner et al, 1999, IB: 1:2,000), respectively, as well as antibodies to the C-terminus of APP (6687, Steiner et al, 2000, IB 1:1,000), and to total Aβ (3552, Yamasaki et al, 2006, immunoprecipitation (IP) 1:500–1:7,500 and 2D8, Shirotani et al, 2007, IB 3 μg/ml), have been described previously. Neoepitope-specific antibody to Val50 of AICD (IB 5 μg/ml) was a gift from Eli Lilly and Company and has been described before (Chavez-Gutierrez et al, 2012). End-specific antibodies to Aβ40, Aβ42, and Aβ43 characterized previously (Saito et al, 2011) were obtained from IBL (JP18580, JP18582, and JP 18583, immunohistochemistry (IHC) 2 μg/ml). Antibodies NT1 to the PS1 NTF (Covance SIG-39194, IB 1:2,000), N1660 to NCT (Sigma N1660, IB 1:1,000), 22C11 to secreted soluble APP (Merck Millipore MAB348, IB 1:5,000), 4G8 to Aβ (Covance SIG-39220, IP 1:500–1:2,500; and BioLegend 800701, IHC 1:500), Y188 to the APP C-terminus (Abcam ab32136, IB 1:4,000), Penta-His (Qiagen 34460, IB 1:2,000), and AT-8 to hyperphosphorylated microtubule-associated protein tau (Thermo Scientific MN1020, IHC 1:200) were obtained from the indicated companies. Rat monoclonal antibody 5E12 (IB 2.5 μg/ml) of IgG2a subclass was raised against residues 313-333 (SKYNAESTERESQDTVAENDD) of human PS1.

### Cell lines

Pooled as well as single-cell clones of HEK293 cells stably co-expressing Swedish mutant APP with WT and mutant PS1

constructs were generated and cultured as described (Steiner *et al*, 2000; Page *et al*, 2008). Culture, transfection, and viral transduction of PS1/2$^{-/-}$ double-knockout MEF cells have been described before (Kuhn *et al*, 2010; Kretner *et al*, 2013).

## Protein analysis

PS1 and PS2, NCT, full-length APP, C-terminal APP fragments, secreted APPs and Aβ were analyzed as described (see Kretner *et al*, 2013 and references therein). Individual Aβ species were quantified by a previously characterized ELISA (Saito *et al*, 2011) using end-specific C-terminal antibodies to Aβ40, Aβ42, and Aβ43 obtained from IBL according to the instructions of the supplier. MS analysis of secreted Aβ species immunoprecipitated with antibody 4G8 was performed as described previously (Page *et al*, 2008) except that samples were analyzed on a 4800 MALDI-TOF/TOF Analyzer (Applied Biosystems/MDS SCIEX). To analyze individual Aβ species by immunoblotting, Tris-Bicine-Urea SDS-PAGE was used (Wiltfang *et al*, 1997). To improve the separation of longer Aβ species, we used a 12% stacking gel containing 4M urea and an 8% separation gel containing 8M urea. Quantitation of bands from immunoblots was done using the LAS-4000 image reader (Fujifilm Life Science) and Multi-Gauge V3.0 software for analysis.

## Cell-free γ-secretase assay

Membrane fractions of PS1/2$^{-/-}$ MEF cells stably transduced with PS1 WT or PS1 L435F were prepared as described (Sastre *et al*, 2001) and solubilized with 1% CHAPSO. Following a clarifying spin (100,000 × *g*, 30 min, 4°C), γ-secretase activity was assessed using recombinant 1.4 μM C100-His$_6$ substrate (Edbauer *et al*, 2003) in assay buffer (150 mM sodium citrate pH 6.4, 0.5 mg/ml phosphatidylcholine, 10 mM DTT, 0.1 mg/ml BSA, 0.25% CHAPSO, 1 × PI) in the presence or absence of 0.5 μM L-685,458 (Merck Millipore). Samples were separated by SDS–PAGE on 10-20% Tris-Tricine gels (Invitrogen), and AICD generation was analyzed by immunoblotting using antibodies Penta-His or anti-AICD Val50.

## Patient samples

Autopsy tissue samples and paraffin sections were provided by the Massachusetts Alzheimer's Disease Research Center. All human autopsy tissue was collected in accordance with the protocol approved by the Institutional Review Board of the Massachusetts General Hospital in accordance with the Ethical Principles and Guidelines for the Protection of Human Subjects of Research (the "Belmont Report") and the requirements of the Health Insurance Portability and Accountability Act (HIPAA) of 1996, as well as applicable regulations. Consent for research use of tissues was obtained from the next of kin of the deceased at the time of death and prior to performance of the autopsy. All cases are listed in Table EV1.

## Mutation analysis

DNA was extracted from frozen tissue samples of all FAD cases using Maxwell System (Promega GmbH, Mannheim, Germany) and quantified by NanoPhotometer® P-Class (Implen GmbH, München, Germany). Two primers (forward primer, 5′-TTGCCTGAAAAT

### The paper explained

#### Problem

The etiology of Alzheimer's disease (AD) is currently explained by two opposing hypotheses. The prevailing and widely accepted amyloid cascade hypothesis states that abnormal accumulation of longer amyloid-β peptide (Aβ) species, such as Aβ42 triggers the disease. An alternative hypothesis, the presenilin hypothesis, states that loss of essential presenilin functions causes the disease and that aberrant Aβ generation is a secondary process. As proteolytic subunit of γ-secretase, presenilin not only generates various Aβ species differing in their C-termini from the amyloid precursor protein (APP) but also cleaves many other crucial substrates such as Notch-1. As a consequence, maintaining rather than inhibiting presenilin function should be beneficial in AD treatment strategies. Thus, clarifying the relevance of the presenilin hypothesis has obvious implications for AD drug development.

#### Results

A major argument for the presenilin hypothesis has been the phenotype of the presenilin-1 (PS1) L435F mutant. This mutant causes a severe loss of function with nearly undetectable Aβ generation, but intriguingly causes familial AD (FAD). To explain this paradox, a transdominant effect of the mutant on PS1 WT was proposed. We now show that, strikingly, the PS1 L435F mutant causes a robust generation of Aβ43, a species, which was not investigated in previous analyses of this mutant. We further show that PS1 L435F also produces Aβ43 in the absence of WT presenilin excluding a dominant-negative mechanism of this mutant. Finally, unlike in controls, neuropathological analysis of the PS1 L435F FAD cases revealed a robust deposition of Aβ43 in postmortem brain tissues.

#### Impact

The finding that the PS1 L435F mutant preferentially generates the previously overlooked highly amyloidogenic Aβ43 by its intrinsic residual activity supports the concept of the amyloid cascade hypothesis. Accordingly, loss of essential presenilin functions is unlikely to be the primary trigger of AD. Thus, disease-modifying approaches for AD aiming at selectively targeting the generation and/or clearance of amyloidogenic Aβ species should remain a major focus to develop therapeutic interventions to AD.

GCTTTCATAATTAT-3′; reverse primer, 5′-GGAATGCTAATTGGTC-CATAAAAG-3′) were designed to amplify a 199-bp product flanking the hot spot mutation L435F in the exon 12 of PS1. DNA was amplified by PCR using the Multiplex PCR Kit (Qiagen, Hilden, Germany) following manufacturer's instructions. PCRs were performed in a total volume of 25 μl with 20–30 ng genomic DNA. The PCR mixture was performed with an initial denaturation for 15 min at 95°C, cycled 38 times (94°C for 30 s, 54°C for 60 s, and 72°C for 120 s) and 72°C for 10 min for a final extension. Visualization of the PCR product was performed by gel electrophoresis. PCR product was purified with the DNA Clean & Concentrator Kit (Zymo Research Europe GmbH, Freiburg, Germany) and directly sequenced on an ABI 3130 Genetic Analyser (Applied Biosystem, CA) using the same primers as above.

## Histological stains

Hematoxylin–eosin stains and Gallyas silver stains were performed according to standard protocols.

## Immunohistochemistry

Deparaffinized sections were pretreated with 90% formic acid for 5 min before incubation with rabbit polyclonal antibodies specific to Aβ40, Aβ42, or Aβ43 or with mouse monoclonal antibody 4G8 recognizing all Aβ isoforms. For the detection of hyperphosphorylated microtubuli-associated protein tau, sections were microwaved before incubation with mouse monoclonal antibody AT-8. Immunohistochemistry was performed with a Ventana BenchMark using the i-view DAB detection kit.

Expanded View for this article is available online.

## Acknowledgements

We are grateful to Bart De Strooper, Ralph A. Nixon, Alison Goate, and Paul Szekeres for reagents; Matthew Frosch and Brad Hyman for providing paraffin sections and frozen brain tissue samples; and Alice Sülzen, Brigitte Kraft, and Michael Ruiter for technical assistance. We also thank Axel Imhof for giving access to the mass spectrometer of the Protein Analysis Unit of the Ludwig-Maximilians-University Munich and Ignasi Forné, Pierre Schilcher, and Andreas Schmidt for technical help with MS analysis. This work was supported by the Friedrich-Baur-Stiftung (HS), the Deutsche Forschungsgemeinschaft (FOR2290) (HS and SFL), the Breuer Foundation Alzheimer Award (SFL), an advanced grant of the European Research Council under the European Union's Seventh Framework Program (FP7/2007–2013)/ERC Grant Agreement No. 321366-Amyloid (CH) and the general legacy of Mrs. Ammer (to the LMU/chair of CH).

## Author contributions

BK, JT, CH, and HS conceived and designed experiments. BK and JT performed biochemical experiments. AF performed mass spectrometry analysis. TA performed and evaluated histological and immunohistochemical stains and documented their results. JM and AG were responsible for DNA sequence analysis. PK and SFL generated stably transduced MEF cells. EK generated monoclonal antibody 5E12. BK, JT, AF, CH, TA, and HS analyzed data and interpreted results. HS supervised the project and wrote the paper with contributions from BK, JT, JM, and TA.

## Conflict of interest

The authors declare that they have no conflict of interest.

## For more information

http://www.alzforum.org/news/research-news/mutant-presenilin-knock-mice-mimic-knockouts-stir-old-debate: A controversial discussion whether loss of presenilin function is at the core of AD after it was reported that PS1 L435F knockin mice display cognitive decline and neurodegeneration (Xia *et al*, 2015).

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
