## [Review Process File · EMBO Molecular Medicine]

Generation and deposition of A β 43 by the virtually inactive presenilin-1 L435F mutant contradicts the presenilin loss of function hypothesis of Alzheimer's disease

Benedikt Kretner, Johannes Trambauer, Akio Fukumori, Janina Mielke, Peer-Hendrik Kuhn, Elisabeth Kremmer, Armin Giese, Stefan F. Lichtenthaler, Christian Haass, Thomas Arzberger, Harald Steiner

Corresponding author: Harald Steiner, Biomedical Centre, Metabolic Biochemistry, Ludwig-Maximilians-University Munich & DZNE German Center of Neurodegenerative Diseases

Review timeline:	Submission date:	16 October 2015
	Editorial Decision:	05 November 2015
	Revision received:	05 February 2016
	Acceptance:	25 February 2016
	Accepted:	25 February 2016

Transaction Report:

Editor: Céline Carret

1st Editorial Decision	05 November 2015
------------------

Thank you for the submission of your manuscript to EMBO Molecular Medicine. We have now heard back from the two referees whom we asked to evaluate your manuscript. You will see from the reports copied below that both referees are clearly enthusiastic and recommend publication. However, before to proceed, there are a few minor issues that need to be addressed.

In addition to the referees comments, we also would like to encourage you to fulfil all editorial requirements at this stage in order to proceed faster while the revision will be evaluated (see below).

We would welcome the submission of a revised version for further consideration and depending on the nature of the revisions, this may be sent back to the referees for another round of review.

I look forward to seeing a revised form of your manuscript as soon as possible.

**** Reviewer's comments ****

Referee #1 (Comments on Novelty/Model System):

The authors first use cell culture systems to unequivocally demonstrate that L435F mutant PS1 does indeed produce Abeta peptide and specifically the 43-residue peptide (Abeta43). In particular, use of PS1/PS2 double-knockout cells transduced with PS1 L435F definitively shows that this mutant is not producing Abeta by trans interaction with wild-type presenilin. The authors also use brain sections from Alzheimer subjects to show that the L435 PS1 mutant carriers deposited amyloid plaques composed of Abeta43.

Referee #1 (Remarks):

This is a short but definitive and important study that counters the notion that presenilin complete loss-of-function is responsible for Alzheimer's disease (AD) pathogenesis. The L435F PS1 mutation has been recently touted as a mutation without any proteolytic function but found to cause dominantly inherited early-onset familial AD. In this "presenilin hypothesis", mutant presenilin is thought to interact with wild-type presenilin to alter the latter's gamma-secretase activity to elevate the ratio of Abeta42 to Abeta40. Kretner and colleagues now show that L435F PS1 does indeed have proteolytic activity of its own, independent of any interaction with wild-type presenilins (using PS1/PS2 double knockout MEFs), but that what is produced is primarily Abeta43. Abeta43, like Abeta42, is also highly aggregation-prone, and indeed, the authors show that brain sections from PS1 L435F familial AD subjects have Abeta43-containing amyloid plaques, similar to what has been previously reported with another PS1 mutation, R278I.

It's high time the "presenilin hypothesis" was put to rest. We already know what complete loss of function of presenilin or other gamma-secretase components do in human population: they cause a severe skin disease, not neurodegeneration (Wang B et al. Gamma-secretase gene mutations in familial acne inversa, *Science*, 2010). And the idea that mutant PS1 can alter Abeta in trans through the wild-type protein is based on a poorly executed and interpreted study in which detergent was used above its critical micelle concentration to co-immunoprecipitate mutant PS1 with wild-type (see ref 11 in manuscript). Nevertheless, the idea has gained considerable attention and generated confusion and misconceptions in the field and even in the public at large. This new study conclusively excludes PS1 L435F as the poster child for the presenilin hypothesis.

This manuscript is completely sound technically and very well written. I only suggest the following minor changes: On page 5, please note that the PS1 R278I mouse model was knock-in. Also note that the amyloid deposition seen with the hemizygous knock-in was upon crossing with APP transgenic mice.

Referee #2 (Comments on Novelty/Model System):

Used both cell lines PD1/@ double KO and human tissue.

Referee #2 (Remarks):

The manuscript demonstrates that one presenilin mutant L435F is not a true loss of function mutant but a partial loss of function that appears to decrease both initial cleavage (though not directly shown here) and subsequent processivity. This data along with other recent data largely refute the presenilin hypothesis that has received prominent attention in some journals. This is an important study that will be well cited and is needed to counterbalance those who continue to argue that such mutations show that loss of gamma-secretase activity plays a big role in AD. However, there are some concerns that should be addressed before publication.

1. The statement that Abeta43 is more toxic than Abeta needs to be qualified. All toxicity assays are very artificial. Similarly the report that Abeta43 is an early and preferentially deposited species in AD and mouse models is not universally supported by all studies (see for example several Mass spec studies such as Moore et al ART 2012)
2. The mutation L435F is referred to as extraordinary, I am not sure this is the best adjective.
3. APP CTF accumulate with this mutant but do they accumulate in cells expressing endogenous levels of APP or is this an artifact of APP over expression? I also think showing effects on AICD production and cleavage site utilization are important.
4. The mass spec needs to have some data on the expected and observed MW of the Abeta42 and 43 peaks. Please provide data that Abeta 42 and 43 synthetic peptides fly differentially in the MS used for these studies. These data are not congruous with the Western blots and can be explained by differential efficiency of flying the 42 versus 43 peptides. Also are the shifts shorter Abeta peptides reproducible in multiple spectra?
5. The ELISA data is presented in a way that is a little misleading. (Figure 1g) . Please show absolute levels.
6. Please show standards for migration of Abeta on the gels in each panel.
7. The term massive Abeta43 deposition is used to describe the PS1 L435F cases. I think this is a little overstatement without quantitative assessment based on biochemistry. The amount of Abeta detected by IHC can vary depending on antibody, fixation and antibody retrieval. Just state there is a lot more Abeta43 without such hyperbole. Indeed, in general I think the manuscript can be toned down a bit and just state the data as it stands.

In our revised version, we have addressed the points raised by the two reviewers, which we felt, like you, were both very enthusiastic about our work and recommended publication. Note that in addition to your editorial requests, a longer abstract has been written, and the number and set up of the main and extended version figures have changed at various positions. We also included data confirming the mutational status of the PS1 L435F cases, which is supplied in the new EV Fig 2.

Please find below our point-to-point response to the reviewer comments:

Reviewer 1:

I only suggest the following minor changes: On page 5, please note that the PS1 R278I mouse model was knock-in. Also note that the amyloid deposition seen with the hemizygous knock-in was upon crossing with APP transgenic mice.

Response:

The requested details on the PS1 R278I mouse model have been added.

Reviewer 2:

1. The statement that Aβ43 is more toxic than Aβ42 needs to be qualified. All toxicity assays are very artificial. Similarly the report that Aβ43 is an early and preferentially deposited species in AD and mouse models is not universally supported by all studies (see for example several Mass spec studies such as Moore et al ART 2012)

Response:

We based our reasoning on toxicity and amyloidogenicity of Aβ43 mostly on the evidence presented in the Nat. Neuroci. 2013 report by Saito et al. but agree with the reviewer that after reassessing the literature, a more balanced account on these properties of Aβ43 as compared to Aβ42 is warranted. We have thus rewritten the respective part in the manuscript and included additional citations in our revised version as follows:

“Although Aβ43 is secreted only in very minor amounts under physiological conditions (Page et al, 2008; Saito et al, 2011) it is found in Aβ plaques in sporadic and FAD brain (Iizuka et al, 1995; Parvathy et al, 2001; Saito et al, 2011; Sandebring et al, 2013; Welander et al, 2009). Like Aβ42, it is highly neurotoxic, although the extent of toxicity compared to Aβ42 varies among assays (Burnouf et al, 2015; Meng et al, 2015; Saito et al, 2011). Aβ43 is deposited earlier than Aβ42 in mouse models of AD indicating that Aβ43 is a potent and probably also a primary nucleation factor of Aβ aggregates in vivo (Saito et al, 2011; Zou et al, 2013).”

2. The mutation L435F is referred to as extraordinary, I am not sure this is the best adjective.

Response:

We exchanged “extraordinary” to “remarkable”.

3. APP CTF accumulate with this mutant but do they accumulate in cells expressing endogenous levels of APP or is this an artifact of APP over expression? I also think showing effects on AICD production and cleavage site utilization are important.

Response:

APP CTFs do not only accumulate with the PS1 L435F mutant in APP overexpression conditions (Fig 1B), but also at endogenous levels of APP expression, as shown in Fig 2A for the analysis of endogenous mouse APP CTFs in MEF cells. As requested by the reviewer, we have included an analysis of AICD generation by the PS1 L435F mutant, which was, as expected, nearly completely blocked. Due to the extremely low AICD amounts, further analysis with respect to ϵ -cleavage site utilization was not possible. However, immunoblotting with a neopeptide-specific AICD antibody directed against Val50 of AICD suggested that at least a portion of the AICD was normally cleaved between Leu49 and Val50 by the mutant γ -secretase. As these data are not central to the study, we show these in the new EV Figs 1A and B.

4. The mass spec needs to have some data on the expected and observed MW of the Abeta42 and 43 peaks. Please provide data that Abeta 42 and 43 synthetic peptides fly differentially in the MS used for these studies. These data are not congruous with the Western blots and can be explained by differential efficiency of flying the 42 versus 43 peptides. Also are the shifts shorter Abeta peptides reproducible in multiple spectra?

Response:

As requested, we have included data of the expected and observed molecular weight of A β 42 and A β 43. This information is provided in the legend of the new Fig 1H.

The reviewer further noted a potential discordance between immunoblot and mass-spectrometry (MS) data (previous Figs 1D and E), which might possibly be attributed to differential flight characteristics of A β 42 and A β 43. We found that A β 42 and A β 43 peptides fly similarly in our hands (Figure for Reviewer). Consistent with this, close reinspection of the data shows that in agreement with the immunoblot data, which show more A β 43 than A β 42, there is also a somewhat higher peak of A β 43 than of A β 42 in the mass spectrum of PS1 L435F. Similarly, there is a higher A β 42 than A β 43 peak for the PS1 L166P mutant, which again matches the immunoblot data.

Figure for Reviewer: Identical amounts of A β 42 and A β 43 standard peptides fly similarly in MS, which is reflected in similar peak heights.

However, we have nevertheless repeated the experiment and now provide a better quality spectrum with a clearly higher A β 43 than A β 42 peak for the PS1 L435F mutant fully resolving this potential ambiguity. Thus, the data of A β 42/43 levels assessed by SDS-PAGE (new Fig 1G) and MS (new Fig 1H) are entirely congruent. We have replaced previous Fig 1E with this new experiment (new Fig 1H).

Finally, the observed shifts to shorter peptides in the mass spectra in the mutant are reproducible and this is a potentially interesting observation. However, as this is beyond the scope of the study we did not comment or elaborate it in our revised version.

5. The ELISA data is presented in a way that is a little misleading. (Figure 1g) . Please show absolute levels.

Response:

Has been done. See new Fig 1F (replacing previous Fig 1G). This new experiment also has a higher n number (n = 6) than the previous one (n = 3).

6. Please show standards for migration of Abeta on the gels in each panel.

Response:

Co-migration with A β 40, 42, and 43 standards is now shown for new Figs 1G and 2B.

7. The term massive Abeta43 deposition is used to describe the PS1 L435F cases. I think this is a little overstatement without quantitative assessment based on biochemistry. The amount of Abeta detected by IHC can vary depending on antibody, fixation and antibody retrieval. Just state there is a lot more Abeta43 without such hyperbole. Indeed, in general I think the manuscript can be toned down a bit and just state the data as it stands.

Response:

We agree with the reviewer that it is difficult to conclude much on the amounts of A β 43 deposited in the patient brains given the mentioned technical issues with IHC. The term “massive” may indeed be suboptimal and thus we changed it to “substantial”. In addition, as requested, we have generally toned the statements down.

Taken together, we believe that we have convincingly addressed all points raised by the reviewers so that our manuscript should be acceptable for publication.

Acceptance

25 February 2016

Please find enclosed the final report on your manuscript. We are very happy to inform you that your manuscript is accepted for publication and is now being sent to our publisher to be included in the next available issue of EMBO Molecular Medicine.

Congratulations on your very interesting work!

***** Reviewer's comments *****

Referee #2 (Remarks):

The authors have largely addressed my concerns. With respect to the issue of the toxicity of Abeta43, the authors might wish to add a sentence or two about what would be the definitive studies.

Corresponding Author Name: Harald Steiner
 Journal Submitted to: EMBO Molecular Medicine
 Manuscript Number: EMM-2015-05952 revised version